# Nervous-Like Circuits in the Ribosome Facts, Hypotheses and Perspectives

**DOI:** 10.3390/ijms20122911

**Published:** 2019-06-14

**Authors:** Youri Timsit, Daniel Bennequin

**Affiliations:** 1Mediterranean Institute of Oceanography UM 110, Aix-Marseille Université, CNRS, IRD, Campus de Luminy, 13288 Marseille, France; 2Institut de Mathématiques de Jussieu - Paris Rive Gauche (IMJ-PRG) Université Paris Diderot, bâtiment Sophie-Germain, 8, place Aurélie Nemours, 75013 Paris, France

**Keywords:** ribosome, translation, network, neuron, nervous circuit, information processing, evolution, complexity, protein interface

## Abstract

In the past few decades, studies on translation have converged towards the metaphor of a “ribosome nanomachine”; they also revealed intriguing ribosome properties challenging this view. Many studies have shown that to perform an accurate protein synthesis in a fluctuating cellular environment, ribosomes sense, transfer information and even make decisions. This complex “behaviour” that goes far beyond the skills of a simple mechanical machine has suggested that the ribosomal protein networks could play a role equivalent to nervous circuits at a molecular scale to enable information transfer and processing during translation. We analyse here the significance of this analogy and establish a preliminary link between two fields: ribosome structure-function studies and the analysis of information processing systems. This cross-disciplinary analysis opens new perspectives about the mechanisms of information transfer and processing in ribosomes and may provide new conceptual frameworks for the understanding of the behaviours of unicellular organisms.

## 1. Introduction

More than twenty years ago, Dennis Bray proposed, on the basis of his analysis of bacterial chemotaxis, the idea that proteins may constitute computational elements in cells [1]. This seminal work suggested that “in unicellular organisms, protein-based circuits act in place of a nervous system to control the behaviour” and that “because of the high degree of interconnection, systems of interacting proteins act as neural networks trained by the evolution to respond appropriately to patterns of extracellular stimuli” [1]. However, he noticed that an important difference with true neural networks is “the wiring of these networks depends on diffusion-limited encounters between molecules and for this and other reasons, they have unique features not found in conventional computer-based neural network”. The recent analysis of r-protein networks in the ribosomes of the three kingdoms [2] updates and further enhances this intriguing hypothesis. First, this study has shown that r-protein networks form complex circuits that differ from most known protein networks, in that they remain physically interconnected. Second, these networks displayed some features of communication networks and an intriguing functional analogy with sensory-motor circuits found in simple organisms. It has been proposed that these networks may play at a molecular scale, a role analogous to a sensory-motor nervous system, to assist and synchronize protein biosynthesis during translation. However, we must be aware that the nerve circuits do not have exactly the same properties that the ribosomal proteins circuits have, even in simple organisms having the most primitive nervous systems that are known. For instance, in *coelenterata,* polyps and *medusae* [3,4], the nervous systems share properties with the most elaborate ones that are not observed in the r-proteins networks: diffuse circuits, autonomy (neurons like to be where they want to be), facilitation (action after repetition), and response mediated frequency. In agreement with the last sentence of D. Bray cited above, we do not know today if these properties have analogues in the r-proteins networks.

We analyse here the significance of this analogy and establish a preliminary link between two fields: ribosome structure-function studies and analysis of information processing systems. This cross-disciplinary analysis opens new perspectives about the mechanisms of information transfer and processing in ribosomes and may provide new conceptual frameworks for the understanding of the behaviours of unicellular organisms.

## 2. Facts and Current Paradigms

### 2.1. An Extensive Flow of Information

During the past decades, structural studies of ribosomes have brought a considerable amount of information about the mechanisms of translation [5,6,7,8,9,10,11]. These studies have converged towards a metaphor in which the ribosome is considered a “Brownian nanomachine” that accurately performs mRNA coded protein biosynthesis. However, while these studies have provided a detailed mechanistic understanding of almost all of the translation steps, they simultaneously revealed intriguing ribosome properties challenging this point of view. Indeed, they progressively revealed that many of the ribosome functional properties go far beyond the skills of a simple mechanical machine.

Many experimental studies have shown that—as well summarized in the title of paper of Rhodin and Dinman—“*an extensive network of information flow*” through the ribosome during protein biosynthesis [12]. First, growing evidence indicates that ribosome functional sites (RNA binding sites, decoding centre, peptidyl transferase centre (PTC), peptide exit tunnel) continually exchange and integrate information during the various steps of translation. For example, an allosteric collaboration between elongation factor G and the ribosomal L1 stalk directs tRNA movements during translation [13,14]. In addition, the overall ribosome dynamics are modulated by the three tRNA site occupancies and their aminoacylation status [15]. Additionally, several studies have demonstrated long-range signalling between the decoding centre that monitors the correct geometry of the codon-anticodon and other distant sites such as the Sarcin Ricin Loop (SRL) or the E-tRNA site [16,17]. The peptidyl transferase centre (PTC), the large-subunit rRNA active site where peptide bond formation is catalysed, is also a key node of allosteric communication. PTC and the A-site communicate and are coordinated through the universal protein uL3 [18]. Moreover, specific r-proteins of the ribosomal tunnel play an active role in the translation regulation or co-translational folding by sensing the nature of the nascent peptide and communicating to the PTC or the exit sites [19,20,21]. In addition, recent studies have extended the scope of ribosome sensing systems to a higher level in describing the molecular mechanisms of a quality sensor of collided ribosomes in eukaryotes [22] and showed that sensing may also involve higher-order ribosome architectures to monitor the translation status.

### 2.2. Ribosome Choreography during Protein Biosynthesis

Second, to perform the biosynthesis of proteins, the ribosome must synchronize extremely complex movements by combining small and large-scale motions such as the ratchet-like motion between the two subunits [23]. More than 21 hinges have been identified within the rRNAs for accomplishing independent unit motions in bacterial ribosome [24,25]. Could Brownian molecular motions be efficient and fast for a system with so many freedom degrees without control? Brownian motion is convincingly suggested to be at the ambient origin of the process driving the basic motions of the ribosome [26,27], then rotation and elongation can be thermally driven. It has also been suggested that an out of equilibrium stochastic process, a variant of Totally Asymmetric Exclusion Process (TASEP), can describe approximately the basic sequence of ribosome’s movements [28]. However, this sequence is certainly of a more elaborate nature than a pure TASEP, needing the intervention of external factors for regulation and also an internal control and adaptation, probably helped by the rRNA, mRNA, tRNA and the r-proteins. Interestingly, remote communication processes have been recently shown to participate in the coordination of complex ribosomal movements during translation [29,30], thus suggesting that ribosomal motions may be helped by external synchronization systems.

### 2.3. Ribosome Heterogeneity and Open Questions

Third, the discovery of ribosomal heterogeneity and specialization [31,32], where differences in ribosomal constituents lead to important functional changes in the function of fluctuating cellular contexts, and significantly expand the possible functional repertoires of the ribosome.

How information transit between distant functional sites and how this information is integrated and processed to enable an optimized ribosome activity in a fluctuating cellular environment remains one of the major challenges in the research on the ribosome [17]. Many studies have already addressed this question and revealed that long-range communication occurs in the ribosome and perform information transfer between remote functional sites. This allosteric communication has been shown to either involve rRNA [30,33] or r-proteins [12,18,29]. However, the vocabulary frequently used in these recent studies implicitly reveals the limits of the machine metaphor: “Sensing”, “communicating” and even “taking decisions” [21] are indeed generally employed for describing autonomous organisms and are rarely used for a machine.

## 3. Hypotheses

### 3.1. Ribosome Behaviour

Peter Moore already called into question the “machine metaphor” and suggested considering the ribosome as a macromolecular “device” rather than a “machine” [34]. In view of the extraordinary ribosome properties, it could be asked today if the ribosome is a device, an “intelligent machine” [35], a “Turing machine” [36] or just an organism instead of an organelle? The later view implies that the ribosome has autonomous “behaviour” in the cell and is fully adapted to perform translation in a fluctuating cellular context. This apparently naïve view has two hypothetical implications. First, it infers that the “ribosome behaviour” would be supported by the equivalent of a nervous system at a molecular scale. Second, in an evolutionary point of view, it would lead to generalizing the concept of a “primordial endosymbiosis” to an early prokaryotic cellular origin. This view fits well with recent papers that propose that the ribosome is a self-replicating intermediate between the RNA world and cellular life and may constitute a missing link in the evolution of life [37,38,39]. These papers have shown indeed that the rRNA contains genetic information that encodes the self-replication machinery, all tRNAs and key ribosomal proteins [37,38,39]. Thus, similar to the mitochondria endosymbiosis in eukaryotic cells [40], RNA/protein organisms like the ribosomes may have joined the cytoplasm of a primitive prokaryotic cell. Cells would have subsequently evolved to protect and optimize pre-existing functions [37,38,39].

Our recent analysis of r-proteins networks formed by tiny but highly conserved contacts between r-protein extensions noticed an intriguing analogy of these networks with nervous circuits [2]. This study proposed that in addition to the previously well-described allosteric mechanisms, r-protein networks could also contribute to transfer and process the “information flow” during translation. This provided an integrated framework that could be based, at least in part, on graph theory, to investigate information transfer and processing in the whole ribosome. This hypothesis proposes to analyse the role of the r-protein network in both motion synchronization and information integration during translation.

D. Bray explicitly suggested the “neuron-protein equivalence” in protein circuits [1]. Following Bray’s metaphor, r-proteins may be functionally comparable to neurons and their networks should, therefore, form nervous-like circuits able to transfer and process information at a molecular scale. We examined here this hypothesis knowing that if a protein behaves as a neuron at a molecular scale, it should have properties similar to a neuron: sensing, transferring and integrating information. R-protein networks would be also expected to display some similarities with an interacting ensemble of nervous circuits systems. We explore these analogies to evaluate their significance and their heuristic power in the understanding of the ribosome’s behaviour.

In fact, the metaphor is perhaps more appropriate and precise with the substructures of a CNS, for instance, a collection of neuronal areas. This is indicated in particular by the evolution of eukaryotic ribosomes: although there is a multiplication by ten of the number of bases in rRNA between *C. elegans* and primates and a corresponding growing complexity of the elements of the proteins [41], there are no noticeable changes of their r-protein networks. In comparison, during evolution, the graph of the neuronal networks becomes more and more complex, the numbers of neurons and of their mutual connexions explode. However, in vertebrates, for instance, the whole plan of the CNS is relatively stable. Thus, a better analogy seems to occur between the r-proteins network and interconnected areas of the nervous central system (CNS). The neuronal assemblies become more complex as the individual proteins and their contacts do.

Furthermore, even considering the whole nervous system instead of neurons, we must underline a big difference between the two networks: the r-proteins one has other additional functions than the sensorimotor one or the information transfer, it also has, at the same time, a function of dynamic stabilization or for helping substrate stabilization and discrimination [42,43], but neuronal networks do not have such additional functions. Therefore, when we speak of a metaphor or analogy between the r-proteins systems and the nervous systems, we must remember that this concerns only a part of the function of r-proteins systems.

### 3.2. The r-Protein–Neuron Equivalence

If a protein “equals” or may behave as a neuron at a molecular scale, it should have properties similar to a neuron: sensing, transferring and integrating information. To examine this assumption, we need to summarize some general properties of neurons. Even if there exists an incredible variety of neurons, they have common properties that, taken all together, distinguish them from other kinds of cells [44,45]. They are excitable cells: an electric potential is changing under the action of electromagnetic input, or chemical reactions. Neurons are polarized in general but could in some cases transmit impulse in both directions and synapses can be dendro-dendritic or axo-axonal. The internal potential of neurons may change under the actions of other cells (that can be neuron). Moreover, these changes induce, after propagation (conduction of excitation or inhibition) and thresholding, a change in the state of other cells, for instance, muscles fibres or neurons. This is triggered by the emission of small molecules outside or by direct electric contact, which happens in general at specific loci of the membrane, forming synapses. The cytoskeleton can also change in time in a functional dependent manner.

The basic neuronal circuit, probably the most primitive one, is a reflex arc. A transductor acts on a neuron, which acts on other neurons and so on, in an arborescent manner (roots are the entry and leaves are the end), until the last ending neurons act on muscles to induce a motion or a reaction. This can induce, for example, a contraction of the animal’s body. However, this simple tree circuit is rare because neurons have a tendency to send feedback (directly or indirectly) and they also have ascending (bypassing) feedforward contact as well. Thus, *loops* are the rule. Neurons are almost always present in collections of similarly structured and inter-connected neurons, but the collective behaviours of these collections also rely on neurons in other collections, in different epithelia or areas.

Neurons and neuronal systems have characteristics that we hope to find in the r-proteins system: (1) the activity of individual units (as one brain’s area, one neuron, or one synapse) can be directly correlated to particular states or events in the external world (e.g., the receptive fields (RFs) of principal neurons in the visual or auditory neocortex, in the Thalamus or in the Hippocampus of mammals); (2) however, much more information is contained in the time sequences of activities, in assemblies of units, as sub-populations of neurons or the related areas of the brain [46]; for instance synchronized neuronal populations can represent the memory of specific events and are able to learn new events; (3) the function of neurons (individually or collectively) is the transmission of information and the preparation of actions; this implies that their activities change for adaptation as soon as the mutual information quantity or the motor efficiency decreases [47]. Suppressing information sources or motor consequences leads, in general, to the death of neurons; examples are degeneration of afferent vestibular nerves when an inner ear is destroyed, degeneration of moto-neurons when end effectors are damaged, and the rewiring of cortical areas when the natural sensory input is lacking [48,49,50].

### 3.3. Sensing the Ribosomal Functional Sites

In the ribosome, the structural and functional organisations of the individual r-proteins suggest a close analogy with neurons in both the structural and functional points of view. Most ribosomal proteins are composed of a globular domain that is located at the surface of the ribosomal subunits and long filamentous extensions that penetrate deeply into the rRNA core. Some r-proteins are devoted to forming tiny interactions with substrates or products within the ribosome functional sites. A current view is that these interactions contribute to stabilizing the mRNA or tRNA substrates in their correct positions. For example, uL16 and bL27 extensions both reach the PTC in which they are thought to stabilize the correct positioning of the tRNA-A and tRNA-P, respectively [42]. In a similar manner, residues such as proline 45 that adopts a *cis*-peptide conformation and the post-translationally modified aspartate 89 of uS12 (*E. coli* numbering) are in close proximity to the codon-anticodon helix at the A-tRNA site and are thought to participate in the decoding function [43]. However, it has been also proposed that through their extensions, the r-proteins uL4 and uL22 “sense” the nascent peptide sequence in the exit tunnel of the ribosome [19,20,21].

Sensing the functional sites to monitor the presence, the sequence or the correct orientation of the substrate may be, therefore, generalized to all the ribosomal functional sites. It has been previously noted that the r-proteins “innervate” all the functional sites in a manner similar to a nervous system [2]. It should be noted that the functions of sensing and stabilizing the substrates are not incompatible and may have co-evolved gradually for optimizing ribosome accuracy. Many structural features and the electrostatic nature of the interactions suggest that common electrostatic mechanisms enable them to sense the different actors of translation through transient tiny interactions.

The ribosome actively participates with the correct tRNA selection through the decoding centre [17,51]. However, as noticed by Zaher and Green [17], in contrast to polymerases, the recognition site that monitors the correct codon-anticodon pairing is about 70 Å away from the PTC where the polymerisation takes place. The two sites must, therefore, communicate to perform accurate protein synthesis. During the first step of decoding, the aminoacyl-tRNAs are delivered to the ribosome in a ternary complex with EF-Tu and GTP. The correct codon recognition induces a conformation change in EF-Tu and triggers the GTP hydrolysis leading to the dissociation of the EF-Tu-GDP complex from the ribosome, thus, freeing the aa-tRNA to accommodate into the PTC. Chemical protection experiments and high-resolution ribosome structures have demonstrated that 16S rRNA bases A1492, A1493, G530 and C1518 directly monitor the correct geometry of the codon-anticodon helix formed between the mRNA and the A-tRNA. It has been shown that these rRNA bases relay global structural changes of the small subunit that occur in response to the binding of cognate anticodon stem-loops in the decoding centre. This produces a “closed state” of the small subunit that, in turn, activates the EF-Tu GTP hydrolysis. Both biochemical and structural studies have demonstrated that uS12, uS3 and the interaction between uS4 and uS5 also play a key role in this process [17,51,52]. Figure 1 shows that these r-proteins closely approach the decoding centre through electrostatic interactions involving conserved amino acids. Their side chains may, therefore, sense and transmit the local structural and electrostatic changes associated with the correct tRNA-mRNA pairing.

Figure 2A also shows how uS13 delicately touches the anticodon loop of the tRNA-P with its C-terminal extension and Figure 2B shows how the r-proteins bL35 and bL33 interact with the tRNA-E in the E-site [53] with a few charged amino acids.

Figure 3 shows that sensing the PTC also involves the extensions of the two universal and essential r-proteins uL2 and uL3 [18,54,55,56] approached in a pseudosymmetric manner the two uracil bases U2585 and U2506 (*E. coli* numbering) that move when the PTC is occupied by the CCA end of a tRNA [57,58]. The close proximity of the charged residues of uL2 and uL3 from the heart of the PTC provides a way to electrostatically sense its different conformations and, therefore, to monitor the steps of the translocation.

In the peptide tunnel, which also plays an active role in translation regulation and co-translational protein folding [19,20,21], a similar situation is observed. The tunnel interior is also “innervated” by several extensions including the universal uL4, uL22 and other r-proteins according to the phylum. Figure 4 displays the specific interactions formed between the ribosome stalling peptide *VemP* and the tunnel components of *E. coli* ribosome [59] (pdb 5NWY). Similar to the other functional sites, tiny contacts also involve charged and aromatic residues that are transiently in contact with nascent peptide.

Sensory-proteins may also monitor the rotational states of the two subunits. For example, a hypothesis is that the two different contacts occurring between uL5 (LSU) and uS13 (SSU) may also contribute to detecting the rotated and un-rotated state of the subunits (Figure 5). While the two different contacts involving conserved residues in both proteins provide a way to stabilize the two rotational states of the ribosome, the switch between the two contact types could also be a way to monitor and inform both subunits about which rotational state each is in. Thus, particular r-proteins may both sense the presence/absence of molecules in functional sites and also different states of the ribosome itself, much like molecular-scale neurons.

### 3.4. Transferring Information

#### 3.4.1. Molecular Synapses and Wires

In addition to sensing and/or stabilizing the tRNAs and the nascent peptides in the functional sites, r-protein extensions are systematically involved in r-protein contacts and form complex networks [2]. Most of the r-protein interactions display well-defined interfaces that remain stable once the mature ribosome structure has been formed [60]. With an average area of 200 A^2^, which is too tiny to be rationalized in terms of dimer stabilization, these highly phylogenetically conserved interfaces have probably been selected during evolution to play a specific role in inter-protein communication. Their analysis revealed that most of them share aromatic and basic residues involved in cation-π interactions that were interpreted as the “necessary minimum for communication”. Many interfaces also display more complex motifs with a combination of aromatic residues, charged and proline residues [2]. Interestingly, such basic-aromatic amino acid interactions have been also observed along the extensions where they form arrays of intra-molecular interactions (Figure 6). In some cases, RNA bases substitute aromatic residues ensure the continuity of the repetitive motif. It has, therefore, been proposed that these motifs may participate in the communication between r-proteins. For example, the uL13-uL3 interface, which may function as a communication pathway, is universally conserved across the three domains of life, while other such interfaces are replaced by convergence in archaea and eukaryotes (Figure 6).

#### 3.4.2. Molecular Communication

How the information is distributed in the network and between functional sites still remains to be determined. In the “classic” allostery, locally induced conformational changes are propagated to remote protein sites and modulate their properties [61]. Ribosome allostery has been well documented by the Dinman’s group who revealed many communication rRNA and r-protein pathways between remote ribosome functional sites [12,18,29,54]. In an attempt to identify classical allosteric pathways within the r-protein networks, we have systematically compared the r-protein interfaces in the bacterial ribosome structures solved in different functional states. We have looked for conformational differences that could be interpreted as allosteric switches associated with different functional states [2]. However, this thorough structural analysis has not identified clear distinct conformations that could correspond to allosteric switches. Recent high-resolution structures of ribosomes in different functional states that showed clear reorganizations of functional sites confirmed that the r-protein interfaces and extensions adopted a unique structure.

#### 3.4.3. A new Type of Allostery in r-Protein Networks

This study, therefore, suggested that in addition to previously described classical allosteric mechanisms, other types of allosteric transmission may also occur along the r-protein networks. They may either involve the propagation of an electrostatic perturbation or a charge transfer (electrons or protons) along the extensions and between the r-proteins. Due to their particular electrostatic character, the arrays of cation-π interactions observed along and between the r-protein constitute interesting candidates for propagating an electrostatic perturbation. For example, it is likely that an electrostatic perturbation induced by the binding of a tRNA could be propagated along the r-protein wires and distributed in the whole r-protein network (Figure 6). Interestingly, cation-π interactions that are mainly thought to stabilize protein structures [62] also participate in inter-domain communication in proteins [63]. In mediating the interactions between the neurotransmitters and their receptors, they also play a key role in the central nervous system [62]. Although amino acid interactions are generally viewed as stabilization elements for the protein structure, evolutionarily conserved networks of interacting residues in proteins also play a role in allosteric communication [64]. Thus, the highly conserved arrays of basic and aromatic residues in r-protein networks may reflect their role in a new type of allostery. Figure 6 depicts the arrays observed in a universally conserved pathway that connects the PTC to uL13, through the intermediate of the loop of uL3. Interestingly, the mutation W255 observed in yeast interrupts this pathway and produces severe translation defects [65]. The observation that rRNA bases sometimes substitute the aromatic residues also suggests a mechanism in which rRNA and r-proteins may cooperate to form electrostatic switches that open or close communication pathways (Figure 7A).

Supporting our hypothesis, recent studies have proposed the existence of a new type of allostery that does not involve the propagation of a conformational change but that is rather based on altered protein dynamics or order-disorder transition [66,67,68,69]. We have indeed already noticed that some r-proteins display particular electrostatic properties that enable the allosteric coupling of their remote domains [70,71]. Our hypothesis also fits well with a recent study proposing the idea of dielectric allostery [72]. The authors have revealed that ATP binding induces a local electrostatic perturbation in myosin, altering the electrostatic potential in distant regions. More recently, the role of electrostatic perturbation has also been found in allosteric transmission in a PDZ domain [73]. Long-range resonant interactions may also influence protein dynamics over long distances [74]. Allostery involving electron or charge transfer may be also a possibility. While electron transfer is generally used in photosystem and respiratory complex, recent studies have shown that it can participate in the cross-talk between DNA repair enzymes to precisely locate DNA damages in the double helix [75].

In summary, it is likely that communication pathways in ribosomes may combine the multiple repertoires of the allosteric mechanism that may exist in r-proteins (Figure 7B). In addition, it should be noted that the ribosome provides a very particular electrostatic environment where the extensions are immersed in a sea of rRNA negatively charged phosphate groups. Neither electrostatic properties of the proteins or the electron transfer rates in such a medium are well documented. However, it is known, for example, that buried charged amino acids in proteins are much more charged than solvent exposed ones [76]. It could, therefore, be expected that charge transport and the propagation of electrostatic perturbations may have unusual properties in this particular electrostatic context.

### 3.5. Nervous-Like Circuits in the Ribosome?

Through the formal analysis of networks, graph theory has initiated a new era in biology. Network analysis has opened the possibility to study systems of interacting components as a whole and at various scales. It has also provided new conceptual frameworks for analyzing their collective rather individual properties [77,78]. Networks may indeed form complex systems with emergent properties that cannot be resolved with approaches only focusing on the individual functions of their components. However, studying graphs and networks themselves is not sufficient and must be enriched by the physiological knowledge of cells, their signalisation and biochemical interaction. For instance, motifs of neuronal circuits interfere with biological properties of neurons, synapses and neuromodulators [45]. However, particularly interesting is that the structures of the networks may provide insights about their function. Trophic, signalling, protein interaction or metabolic networks may display functional local properties that evolved to provide distinct collective behaviours. For example, in sociology and economics, it is recognized that communication networks often share common properties referred to as a “small world” that allows for rapid communication and information distribution among each node [79]. In addition to the analysis of the network architecture, graph theory may also focus on different possible and convenient measures of the node centralities—the way by which nodes are connected to each other and their connectivity status in the network. For example, some nodes called “hubs” are more connected than the others and in biology, they may play particular functions in interaction protein networks [80]. Modularity or the existence of functional sub-networks that are coordinated for accomplishing sub-functions can be another key feature of biological networks.

Most protein interactomes studied to date are determined by techniques such as the yeast 2 hybrid method that map the complete collection of the transient protein-protein interactions occurring by free diffusion in a cell [81]. Within the ribosome, the r-protein networks differ significantly from these networks because they display tiny but well-defined and permanent interfaces glued into the core of the ribosome. It should be noted that many r-proteins also have broader cellular functions outside the ribosome that involve transient binding to other proteins [39]. However, r-protein networks described here are very different from the protein networks generally described in the literature that are understood in a totally other sense, through dynamic interactions in time. Interestingly, the comparison of the three kingdom ribosome networks has shown that their connectivity significantly grows during evolution and may correlate with increased ribosome potential to regulate complex tasks [2]. In addition, a preliminary analysis of properties of r-protein networks indicated that they form assortative networks with many hubs connected to other hubs and that displayed particular motifs such as the feed-forward loop observed in information processing networks [82]. These premises, as well as the functional analysis of their r-protein networks, suggested their analogy with nervous systems [2]. We analyse here the significance of this analogy.

#### 3.5.1. Number of Nodes, Connectivity and Evolution

The number of nodes and edges of r-protein networks has the same order of magnitude of small sensory-motor networks of animals such as the *chaetognatha* and *prochordata*, such as *ciona* [3]. However, despite the small number of neurons, how these “simple” nervous systems do function is still not well understood. For example, the nervous circuits of ctenophores and cnidarian are composed of about a hundred interconnected neurons. Tunicate tadpole larvae and *C. elegans* nervous circuits are more complex and content about 300 hundred neurons [83]. Interesting, the number of neurons and connections in nervous circuits grow during the evolution and become more connected in the function of the complexity of the behaviour of the animal and the complexity of their tasks.

In a similar manner, the growth of r-protein network nodes and connectivity parallels the task complexities that should be performed during evolution (complex regulation with cellular status, decision making, selection of mRNA to translate, ribosome heterogeneity). A comparison of bacterial and eukaryotic r-protein networks indicates that the eukaryotic one is more connected and that the number of contacts per r-protein has significantly increased (Figure 8A–C) [2]. In addition, the network comparison has revealed the existence of a common universal network that was probably present in LUCA before the radiation of the three kingdoms. This universal network mostly contains connections between r-proteins and functional sites such as the mRNA, the tRNAs and the peptide exit tunnel and is much more developed in the small subunit (Figure 8D). This suggests that the ribosome-like entities proposed to be key intermediates between prebiotic and cellular evolution [37,38,39] already possessed small but elaborate r-protein circuits. The r-protein networks have then been shaped by selective pressure to gradually increase from this common core and to reach the highest connectivity in the eukaryotic ribosomes. Thus, the growth of the r-protein networks parallels and complements the “classical” evolution by accretion where new rRNA and r-proteins have gradually been added around an ancestral universal structure [41].

#### 3.5.2. Functional Organization

A preliminary analysis of the mathematical properties of r-protein networks indicated that they displayed assortative properties, the presence of hubs and the particular motifs [2] (e.g., feed-forward loops) found in information processing networks [82]. A detailed graph theory analysis of these networks is required to further characterize their mathematical properties. It will be also necessary to analyse more complex geometrical structures of dimension higher than one that has been shown recently to better describe neuron networks [84]. Such a study would provide interesting insights about r-protein network evolution.

One of the best analogies between ribosomal networks and simple neuronal networks is their modular structure. Even in simple organisms, as tunicates having a few hundred neurons, but being close to the vertebrates, several distinct sensori-systems work together. For instance, in the adult ascidian *Ciona intestinalis*, a wide range of innervation patterns connects two distinct systems: the siphon and dorsal nervous system [85]. The siphon system is a peripheral sensorimotor system, controlling the oral and atrial siphons with eight siphon lobes and tentacles. It is responsible for food intake and contains, in particular, peptidergic neurons. On the other hand, the dorsal system includes the cerebral ganglion, responsible for absorption, digestion, excretion, reproduction, involving the aCH, GLU, GABA neuromodulators. In addition, there exists a ventral nervous system, innervating the heart, the gill, the stomach, and the endostyle (filter-feeding) [85]. Thus, several networks, made by physiologically different neurons, are interconnected and coordinated.

A similar modular organization is observed in the r-protein networks. Figure 9 shows that in the bacterial ribosome, r-proteins are organized into a few interconnected sub-networks grouped around the functional sites: decoding centre, tRNA sites, factor binding, PTC and tunnel. This r-protein network organization, therefore, suggests that the r-proteins not only sense the functional sites but also collectively share the information that circulates during translation. It is likely that the information about the occupancy/orientation of substrates in the different functional sites is locally integrated into the sub-networks and then transmitted and processed between the different sub-networks (Figure 10).

First, in the SSU, the three A-, P- and E-tRNA sites are interconnected through a quasi-uninterrupted sub-network of sensors and inter-sensor bridges such as uS10 (depicted in blue in Figure 9). Note that the sensor r-proteins (uS3-uS5 or uS7-uS9) are generally interconnected between themselves. This sub-network roughly follows the mRNA path and establishes an uninterrupted communication pathway from the decoding site (A-site) to the E-tRNA site. This circuit is connected to another one which is nearly perpendicular to it and coloured in green. In the green circuit, the r-proteins are candidates to transfer information between the anticodon sensors (uS13) and the elbow sensors (uL5) of the tRNA-P. This sub-circuit also monitors the status of the rotated state of the two subunits (see Figure 3). uS12 distributes the information taken from the decoding centre in a different region of the small subunit through its connection with uS8 and uS17 (orange in Figure 8) and may, therefore, be involved in the large scale motion of the SSU associated with the recognition of the correct codon-anticodon pair in the A-site. Interestingly, the blue and orange sub-networks are joined at the node bS6 that connects the uL2, which is directly connected to the PTC in the large subunit. bS6, therefore, plays a key role in connecting the small subunit networks to the PTC.

Another particularly interesting circuit depicted in red connects the E-site sensors to the tunnel sensor r-proteins uL4, through uL15. These r-proteins may, therefore, contribute to exchanging information between the tunnel and the E-tRNA site and may contribute to synchronizing the speed of the tRNA translocation with the growth of the nascent peptide (see above and Figure 4). On the other hand, particular mRNA sequences are more prone to slippage errors and stalling. To perform an efficient and accurate template protein synthesis, the ribosome must, therefore, continuously exchange information between these two functional sites.

Finally, a densely connected sub-network (yellow) relates the r-proteins (uL11, uL6) that transiently bind translation factors with the peptide tunnel and the PTC. The uL3 node occupies a key position at the crossway of several pathways towards the peptide tunnel sensors. One route that passes through the two intermediates nodes uL13 and bL20 connects uL4. The other one connects uL2 through the intermediate of bL17. This sub-network is also linked to the E-tRNA site through bL21 and uL15.

The sub-network organizations reveal that uL3 and uL2 that both closely approach the PTC (Figure 3) and have distinct ramifications in the ribosome. uL3 mainly relates the PTC to the large subunit network, including the peptide tunnel and the E-site; uL2 is mainly devoted to establishing communication between the PTC and the tRNA sites in the small-subunit. Note that uL3 also connects the small-subunit through bL19 and uL14 (Figure 10).

## 4. Perspectives

In conclusion, our study proposes that the r-protein networks may have an equivalent function to nervous systems at a nanoscale. These molecular systems are proposed to transfer and integrate the information flow that circulates between the remote functional sites of the ribosome to synchronize ribosome movements and to regulate the protein biosynthesis. Thus, r-proteins may collectively integrate the information taken from distinct sites and similar to a nervous circuit, may help to synchronize the correct tRNA recognition, the tRNA translocation and the growth of the nascent peptide. This hypothesis opens new perspectives in ribosome function, in the evolution of complex systems and in biomimetic technological research of nanoscale information transfer and processing. Considering a collective role of r-proteins may stimulate a new conceptual framework for both conceiving new antibiotics and better understanding the origin of ribosomopathies [86]. For example, mutations that impede the communication pathways such as the W255C [65] may have a general role in translation defects and pathologies. Inversely, specifically targeting some pathways in bacterial r-protein networks or sub-networks may help to produce new efficient antibiotics. On the other hand, this study stimulates and further characterizes and compares r-protein networks to understand how they have evolved. This would provide precious insights into the evolution of information processing in living organisms. It may also help to understand the complex behaviours of unicellular organisms that may use similar networks to integrate and respond to external stimuli. Finally, understanding the molecular mechanisms of information transmission and processing would constitute the basis for conceiving new computing nano-devices.

## Figures and Tables

**Figure 1 ijms-20-02911-f001:**
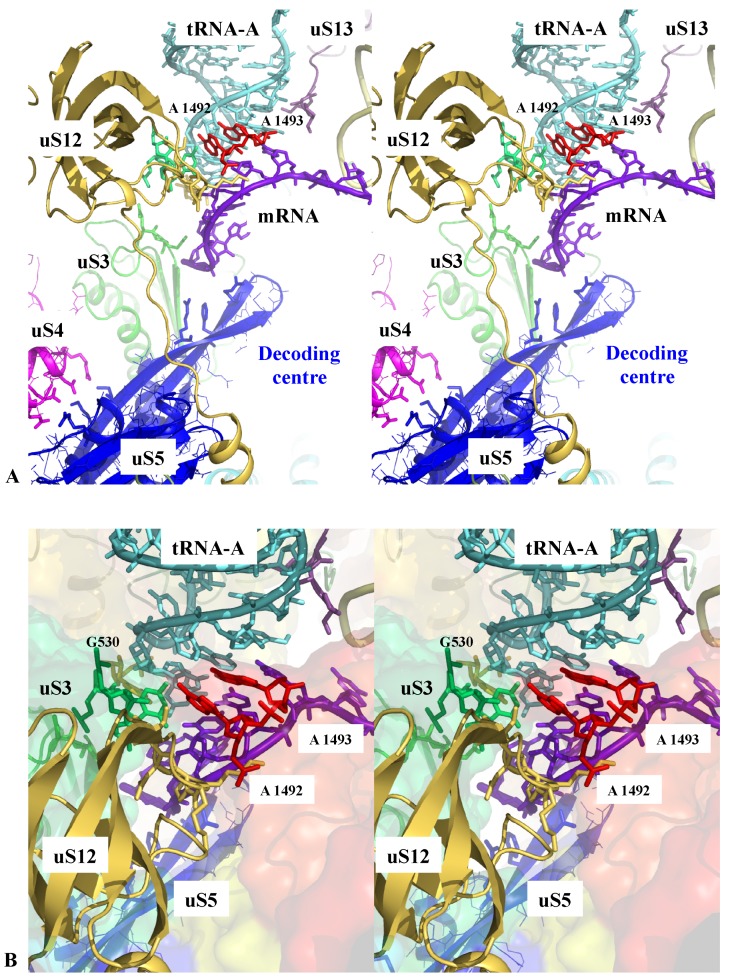
Sensing the decoding centre. Stereoviews of key r-protein and rRNA elements contacting the codon-anticodon interaction between the tRNA-A and the mRNA within the decoding centre. (**A**) overall view; (**B**) detail of the interactions; the rRNA is represented by a transparent surface coloured in the function of the RNA domains (from blue to red in the 5′-3′ direction) (pdb_id: 4y4p).

**Figure 2 ijms-20-02911-f002:**
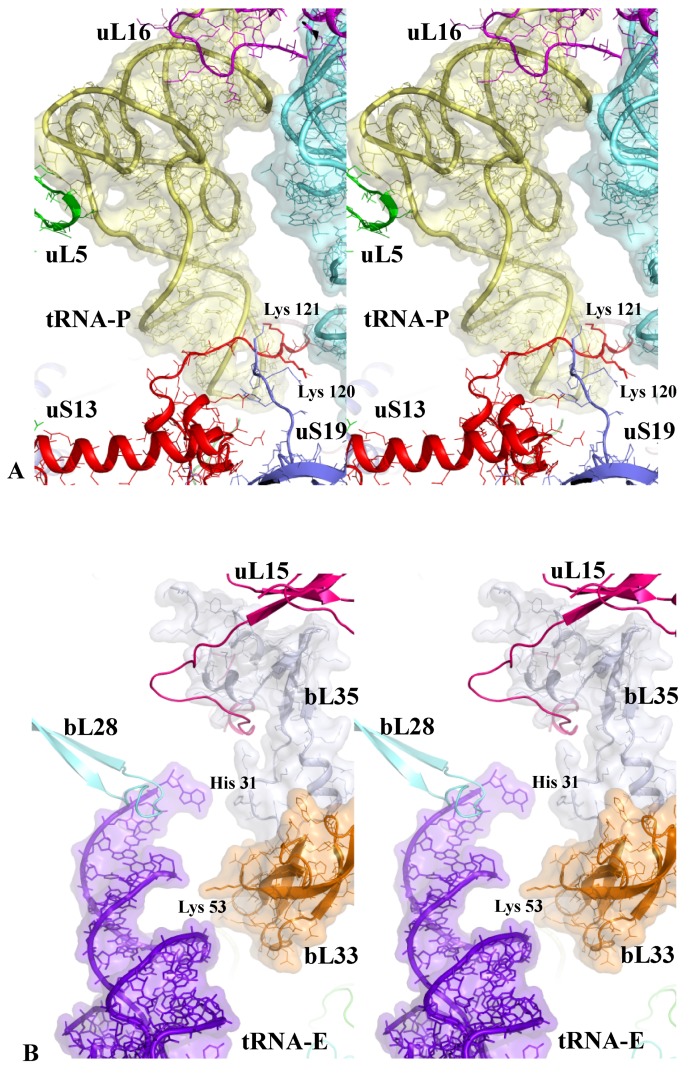
Sensing the P- and E-tRNA sites. Stereoviews of the tiny r-protein tRNA interactions in the (**A**) P- and (**B**) E-tRNA sites (pdb_id: 4y4p).

**Figure 3 ijms-20-02911-f003:**
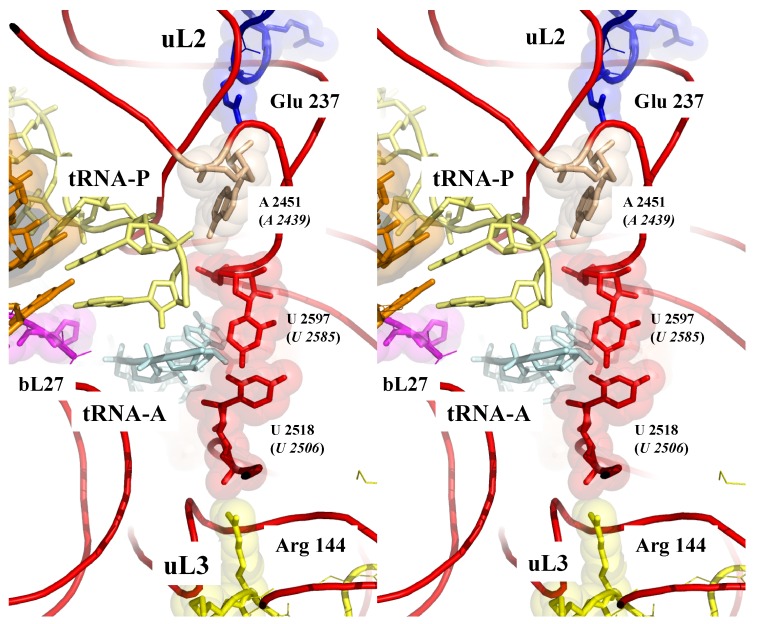
Sensing the peptidyl transferase centre (PTC). Stereoview of the PTC showing the pseudosymmetric interaction of uL2 (top) and uL3 (bottom) with the two uracile bases U 2585 and U 2518 (E. coli numbering) that undergo conformational changes upon tRNA binding. The backbones of the PTC rRNA helices are represented by red cartoons (pdb_id: 4y4p).

**Figure 4 ijms-20-02911-f004:**
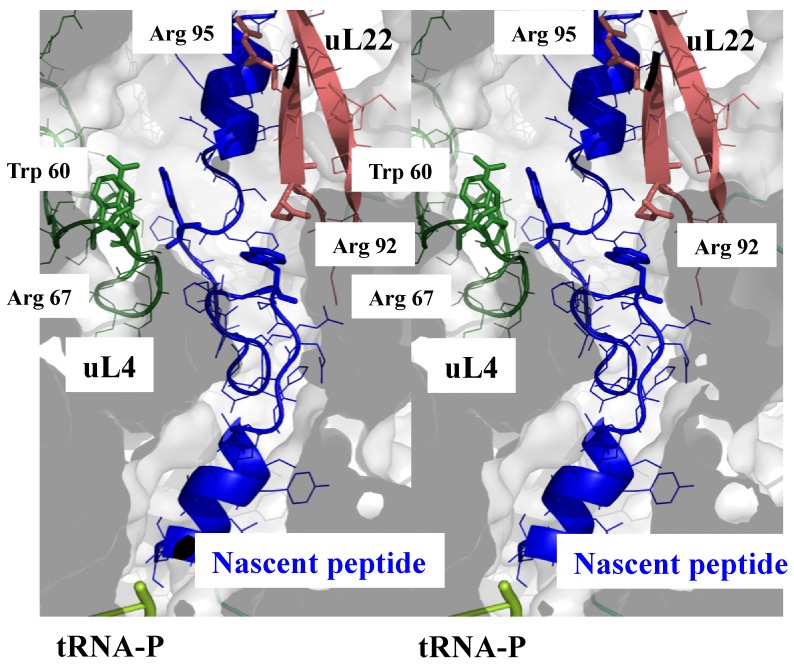
Sensing the interior of the tunnel. Side stereoview of the peptide tunnel showing the interactions between the r-protein uL4 (green) and uL22 (brown) with the stalled nascent peptide *VemP* (blue). rRNA is represented by a white transparent surface (pdb_id: 5nwy).

**Figure 5 ijms-20-02911-f005:**
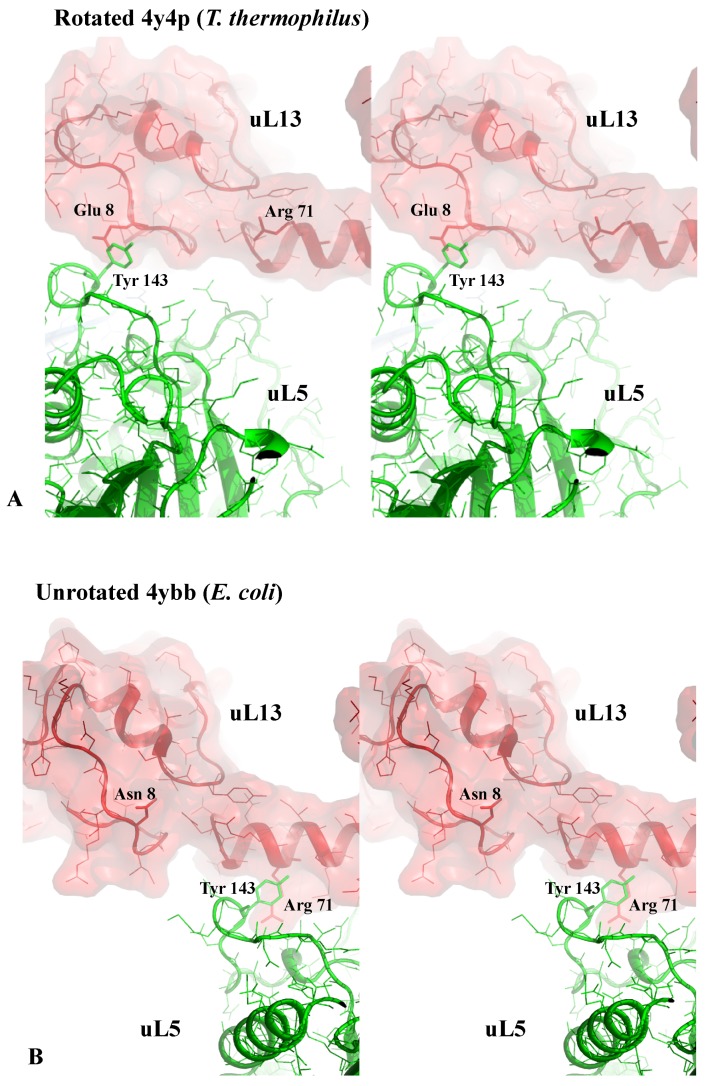
Sensing the rotational state of the two subunits. Stereoviews of the interaction in the inter-subunit bridge formed by uL5 and uS13. (**A**) the rotated state observed in a high-resolution structure of *T. thermophilus* ribosome containing the 3 tRNAs (pdb_id: 4y4p). (**B**) The un-rotated state observed in a high-resolution structure of the *E. coli* ribosome without bound tRNA (pdb_id: 4ybb).

**Figure 6 ijms-20-02911-f006:**
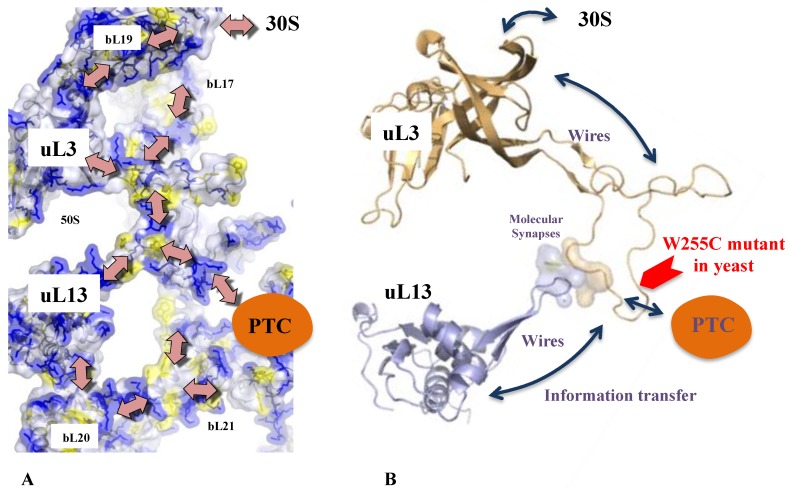
The possible communication pathways along the r-proteins located around the PTC. (**A**) The view of the r-proteins are represented by transparent surfaces coloured according to the amino acids; yellow: aromatic amino acids; blue: basic amino acids. (**B**) The same view without the surface focused on the universal uL3–uL13 interaction. The tiny interface uL3–uL13 is represented by transparent surfaces. The position of the mutant W255C that causes translation defects in yeast is indicated by the red arrow.

**Figure 7 ijms-20-02911-f007:**
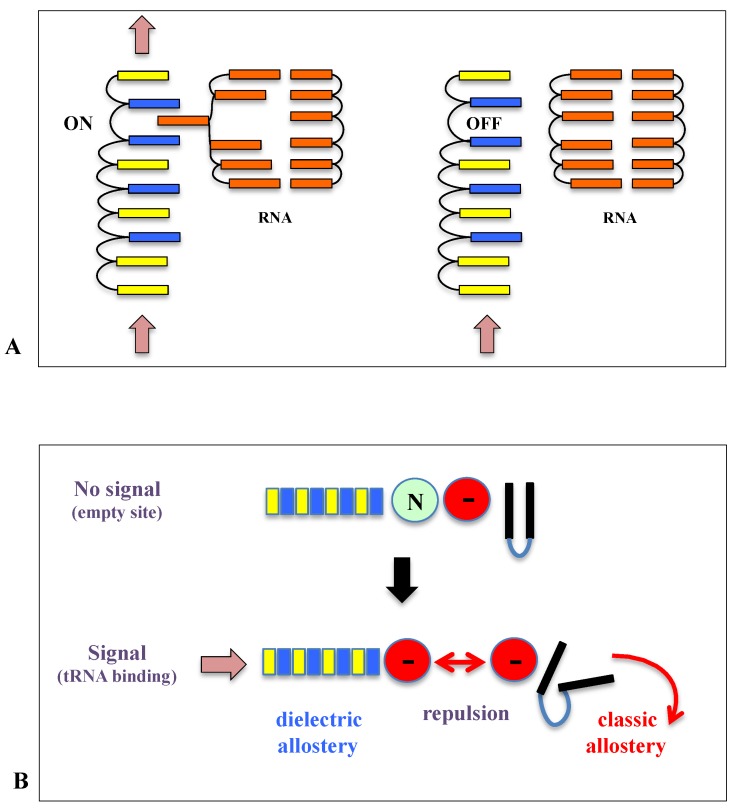
The schematic representation of possible allosteric mechanisms along and between r-proteins. (**A**) The hypothetical mechanism for a cooperative switch formed by the stacking of an rRNA base in the r-protein array of cation-π interactions. Basic (blue), aromatic residues (yellow) or rRNA base (orange) are proposed to transfer charges or electrostatic perturbation (vertical arrow). The signal can be propagated if the rRNA base is stacked in the array and the switch is ON. The signal cannot be propagated if the base is stacked in the rRNA helix and the switch is OFF. (**B**) Summary of the possible combination of several allosteric mechanisms (dielectric and classic). Dielectric allostery propagates an electrostatic perturbation through array of basic (blue) and aromatic (yellow) residues (for example, coming from the binding of tRNA). This may transiently change the charge of a distant neutral residue (N) into a negative one (-) thus inducing a conformational change triggered by a local charge repulsion (or attraction).

**Figure 8 ijms-20-02911-f008:**
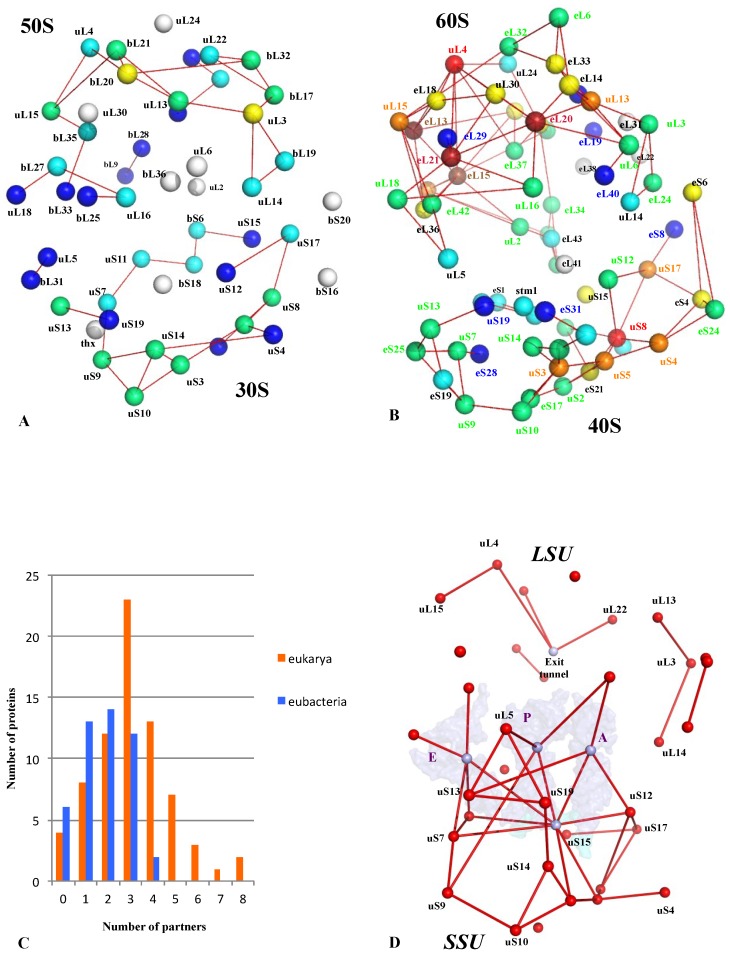
The comparison of bacterial and eukaryotic r-protein networks. Three-dimensional simplified representations of the bacterial (**A**) eukaryotic (**B**) r-protein networks. The centres of mass of the r-proteins (coloured spheres) are connected by red lines. The proteins are coloured in the function of their number of interacting partners. White: 0; blue: 1; cyan: 2; green: 3; yellow: 4; orange: 5; brown: 6; red brick: 7; red: 8. (**C**) 3D simplified representation of the universal network; the three tRNAs and the mRNA are represented by transparent surfaces. (**D**) Histogram reporting the degree distribution (inter-protein contacts) in bacterial (blue) and eukaryotic (orange) ribosome r-protein networks.

**Figure 9 ijms-20-02911-f009:**
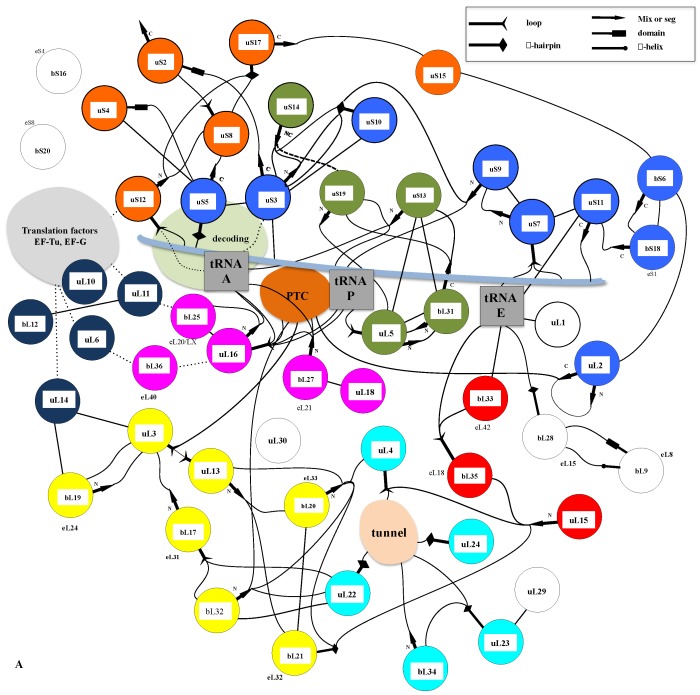
The representation of the bacterial r-protein network. (**A**) A schematic representation of the r-protein nodes (coloured circles), tRNAs (grey squares), mRNA (pale blue line) and the functional centres (coloured ellipsoids) interconnected by black lines that join either the globular domains (circles) or the r-protein extensions (codified by symbols represented in the box). The colours are used to differentiate the functional module or sub-networks. (**B**) Stereoview of the r-proteins and tRNAs in the bacterial ribosome from the A-site tRNA. The modular organisation of r-proteins is shown by colour codes used in (**A**).

**Figure 10 ijms-20-02911-f010:**
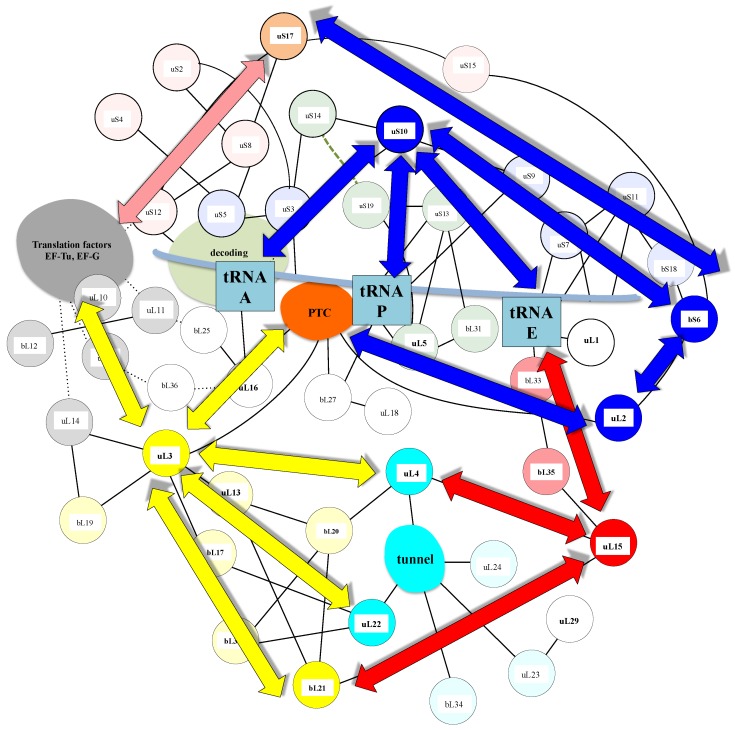
The possible communication pathways depicted by the blue, yellow and red arrays, between the functional sites within the r-protein network.

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
