# Peer review of "Nervous-Like Circuits in the Ribosome Facts, Hypotheses and Perspectives"

_ijms, 2019, doi:10.3390/ijms20122911_

Round 1
Reviewer 1 Report
This is an extremely important paper that proposes an entirely novel approach to understanding ribosomal protein interactions as part of a neuron-like network capable of sensing, information processing, integration and communication. The analogy is well developed, carefully crafted, and provides important new insights into ribosome structure and function. As such, it certainly deserves to be published.
That said, the manuscript needs to be carefully rewritten (see detailed comment below) and might be improved on the three points listed here:
3.1 Ribosome Behavior: On the point about the ribosome possibly being an organism rather than an organelle, the authors might want to consult Root-Bernstein M, Root-Bernstein R. The ribosome as a missing link in the evolution of life. J Theor Biol. 2015 Feb 21;367:130-58. doi: 10.1016/j.jtbi.2014.11.025.
Lines 402-404: “However, the r-protein networks differ significantly from these networks because they display tiny but well-defined and permanent interfaces glued into the core of the ribosome. Thus, r-protein networks described here are very different from the proteins networks generally described in the literature…” I would take exception to this statement. r-proteins are also involved in autologous regulation of their own mRNAs and have broader cellular functions that involve transient binding to other proteins. Root-Bernstein and Root-Bernstein have reviewed the relevant literature in the paper: Root-Bernstein R, Root-Bernstein M. 2018. The Ribosome as a Missing Link in Prebiotic Evolution III: Over-Representation of tRNA- and rRNA-like Sequences and Plieofunctionality of Ribosome-Related Molecules Argues for the Evolution of Primitive Genomes from Ribosomal RNA Modules. Int. J. Mol. Sci. 2019, 20, 140; doi:10.3390/ijms20010140
3.5.2. Functional organization: In this section, is there some reason that there is no summary of an edge and vertex analysis of r-protein interactions that can be compared directly with neuronal systems? Are the networks formed by r-proteins characterized by the same parameters as those of neuronal systems? After all, the power of mathematical analysis is in its ability to abstract out things like network properties so that these can be compared directly without consideration of the components of the systems being compared. So beyond simply being organized in “hubs”, do r-proteins behave like neurons at a mathematical (network) level as described by graph theory parameters?
The paper has many sentences and passages that are difficult to read. Some of these (but certainly not all) are listed below with possible corrections. The authors should re-read their manuscript carefully for other typographical errors and infelicitous sentences!
3.2. Do a protein may “equal” a neuron, at a molecular scale? This heading is not proper English and needs to be fixed!
Lines 180-189: This series of sentences does not make sense as written and does not seem to make a useful point with regard to the broader argument. If it is necessary, the authors need to rewrite it; if it can be eliminated, do so!
“With respect to the environmental changes or to the animal’s actions, it is not exceptional that the 180 activity in time of a unique neuron indicates something significant, for instance receptive fields (RFs) 181 in the neocortex or the Thalamus, or place cells in the Hippocampus, however much more 182 information is obtained from the activity in time of certain convenient sub-populations, called 183 assemblies [43]. This internal “representation” by sequences of synchronized (or poly-synchronized in 184 definite order) supports some kinds of memories and learning, and adaptation. However, everybody 185 thinks that one of the main function of neurons is information transmission, this can be proved in 186 many occasions: for instance if information or motor efficiency (respectively measured by Shannon 187 mutual information or Min-variance execution [44] is loosed due to the wrong activity or lack of 188 activity of some neurons, they change or they die.”
Line 273: “inform both subunits in which rotational state is it.” Should be: “ inform both subunits about which rotational state each is in.”
Line 273-274: “Thus, these few examples propose that similar to neurons at a molecular scale, particular r-proteins may not only sense the presence/absence of molecules in functional sites but also different states of the ribosome itself.” More clear: “Thus, particular r-proteins may both sense the presence/absence of molecules in functional sites and also different states of the ribosome itself, much like molecular-scale neurons.”
Line 284: “In addition to sense and/or stabilize…” Should be: “In addition to sensing and/or stabilizing…”
Lines 287-289: “It has been proposed that with an average area of 200 Å2 and too tiny to be rationalized in terms of dimer stabilisation, these highly phylogenetically conserved interfaces have probably been selected during evolution to play a specific role in inter-protein communication. Their analysis revealed that most of them…” Should be: “With an average area of 200 A2, which is too tiny to be rationalized in terms of dimer stabilization, these highly phylogenetically conserved interfaces have probably been selected during evolution to play a specific role in inter-protein communication. These interfaces…”
Lines 299-301: “For example, the pathway such as the uL13-uL3 interface is universally conserved across the three domains of life, while others are replaced 300 by convergence in archaea and eukaryotes (fig. 6).” Should be: “For example, the 299 uL13-uL3 interface, which may function as a communication pathway, is universally conserved across the three domains of life, while other such interfaces are replaced by convergence in archaea and eukaryotes (fig. 6).”
Lines 370-376: (TYPOS UNDERLINED): “In summary, it is likely that communication pathways in ribosome may combine the multiple 370 repertoires of the allosteric mechanism that may exist in r-proteins (fig. 7B). In addition, it should be 371 noted that he ribosome provides a very particular electrostatic environment where the extensions are 372 immerged IMMERSED? in a sea of rRNA negatively charged phosphate groups. Neither electrostatic properties of 373 the proteins or the electron transfer rates in such a medium are well documented. However it is 374 known for example, that buried charged amino acids in proteins are much more charged that solvent 375 exposed ones [73]. It could be therefore BE expected that charge transport and the propagation of 376 electrostatic perturbations may have unusual properties in this particular electrostatic context.”
Line 434-442: “Even in simple organisms as tunicates having a few hundred of neurons, but close to the 434 vertebrates, several distinct sensori-systems are working together. For instance, in the adult ascidian 435 Ciona intestinalis, a wide range of innervation patterns connects two distinct systems the siphon and 436 dorsal nervous system [81]. The siphon system is a peripheral, sensori-motor system, controlling the 437 oral and atrial siphons, with eight siphon lobes and tentacles. It is responsible of food intake and 438 contains in particular peptidergic neurons. On the other hand, the dorsal system includes the cerebral 439 ganglion, responsible of absorption, digestion, excretion, reproduction, involving aCH, GLU, GABA 440 neuromodulators. In addition there exists a ventral nervous system, innervating the heart, the gill, 441 the stomach, and the endostyle (filter-feeding) [81]. Thus several networks, made by physiologically 442 different neurons, are interconnected and coordinated.” This paragraph needs a new introductory sentence that links it to the next paragraph: something along the lines of “One of the best analogies between ribosomal networks and simple neuronal networks is their modular structure. Even simple organisms such as tunicates, which are evolutionarily close to vertebrates, network several distinct, modular sensory systems together. For instance…”
Line 501: “to nervous systems at a nanoscale. These molecular systems are supposed PROPOSED to transfer and integrate…” [“SUPPOSED” HAS TWO MEANINGS IN COLLOQUIAL ENGLISH THAT CONFUSE THIS SENTENCE: PROPOSE SEEMS TO FIT YOUR MEANING BET
Author Response
Reviewer 1
This is an extremely important paper that proposes an entirely novel approach to understanding ribosomal protein interactions as part of a neuron-like network capable of sensing, information processing, integration and communication. The analogy is well developed, carefully crafted, and provides important new insights into ribosome structure and function. As such, it certainly deserves to be published.
That said, the manuscript needs to be carefully rewritten (see detailed comment below) and might be improved on the three points listed here:
3.1 Ribosome Behavior: On the point about the ribosome possibly being an organism rather than an organelle, the authors might want to consult Root-Bernstein M, Root-Bernstein R. The ribosome as a missing link in the evolution of life. J Theor Biol. 2015 Feb 21;367:130-58. doi: 10.1016/j.jtbi.2014.11.025.
Lines 402-404: “However, the r-protein networks differ significantly from these networks because they display tiny but well-defined and permanent interfaces glued into the core of the ribosome. Thus, r-protein networks described here are very different from the proteins networks generally described in the literature…” I would take exception to this statement. r-proteins are also involved in autologous regulation of their own mRNAs and have broader cellular functions that involve transient binding to other proteins. Root-Bernstein and Root-Bernstein have reviewed the relevant literature in the paper: Root-Bernstein R, Root-Bernstein M. 2018. The Ribosome as a Missing Link in Prebiotic Evolution III: Over-Representation of tRNA- and rRNA-like Sequences and Plieofunctionality of Ribosome-Related Molecules Argues for the Evolution of Primitive Genomes from Ribosomal RNA Modules. Int. J. Mol. Sci. 2019, 20, 140; doi:10.3390/ijms20010140
Many thanks for the references of these very interesting papers (I, II and III) !! We have included and discussed all of them in the present paper that clearly converges with these views of an autonomous ribosome (references 37, 38 and 39 in the new version).
3.5.2. Functional organization: In this section, is there some reason that there is no summary of an edge and vertex analysis of r-protein interactions that can be compared directly with neuronal systems? Are the networks formed by r-proteins characterized by the same parameters as those of neuronal systems? After all, the power of mathematical analysis is in its ability to abstract out things like network properties so that these can be compared directly without consideration of the components of the systems being compared. So beyond simply being organized in “hubs”, do r-proteins behave like neurons at a mathematical (network) level as described by graph theory parameters?
This is a fundamental question that we want to explore in depth in the immediate future. However this question is more complex than it appears at first sight:
1) the suggestion of specific motifs in neuronal networks (Milo,.., Alon Science 2002) was not followed by spectacular progresses;
2) recent explorations in large scale confirmed the existence of cliques and cavities that are better described by higher dimensional objects (complexes) than graphs (cf. Reimann, et al 2017) (reference 84 in the new version)
3) as said in our article, the link between the graphs topology and the functions of neuronal networks depends on chemical and physiological properties of the neurons and synapses that are involved. Thus it seems that graph theory is not sufficient for determining clear properties of neurons networks. Probably the same thing is true for r-proteins networks. Of course this doesn’t forbid to research common motives in graphs, or complexes, or chemically enriched structures, for r-proteins and neurons. A further problem comes from the fact we also mentioned in the article, that the best comparison can hold between connected neuronal areas and r-proteins, not by neurons and r-proteins, which implies that we need not only the structure of the graph made by proteins but also the structure of the multiple connections between two or more proteins, to compare with neuronal networks.
The paper has many sentences and passages that are difficult to read. Some of these (but certainly not all) are listed below with possible corrections. The authors should re-read their manuscript carefully for other typographical errors and infelicitous sentences!
3.2. Do a protein may “equal” a neuron, at a molecular scale? This heading is not proper English and needs to be fixed!
The sentences have been corrected as suggested by reviewer 1
Lines 180-189: This series of sentences does not make sense as written and does not seem to make a useful point with regard to the broader argument. If it is necessary, the authors need to rewrite it; if it can be eliminated, do so!
“With respect to the environmental changes or to the animal’s actions, it is not exceptional that the 180 activity in time of a unique neuron indicates something significant, for instance receptive fields (RFs) 181 in the neocortex or the Thalamus, or place cells in the Hippocampus, however much more 182 information is obtained from the activity in time of certain convenient sub-populations, called 183 assemblies [43]. This internal “representation” by sequences of synchronized (or poly-synchronized in 184 definite order) supports some kinds of memories and learning, and adaptation. However, everybody 185 thinks that one of the main function of neurons is information transmission, this can be proved in 186 many occasions: for instance if information or motor efficiency (respectively measured by Shannon 187 mutual information or Min-variance execution [44] is loosed due to the wrong activity or lack of 188 activity of some neurons, they change or they die.”
The goal of this paragraph was to insist on characteristics of neurons and neuronal systems that we hope to find in the r-proteins system: 1) at the level of individual units (one neuron, one synapse) some elements or events in the external world directly influence the activity, 2) neuronal populations have significance that individual neurons don’t have, 3) the fact that neurons (individually or collectively) have for function the transmission of information and/or the control of movement (or more generally of actions), can be seen in the change which is observed as soon as the mutual information quantity or the motor efficiency decreases. You are right that the manner this was described was too compressed and very badly expressed. We have corrected the paragraph in the new version (lines 180-202)
Line 273: “inform both subunits in which rotational state is it.” Should be: “ inform both subunits about which rotational state each is in.”
Line 273-274: “Thus, these few examples propose that similar to neurons at a molecular scale, particular r-proteins may not only sense the presence/absence of molecules in functional sites but also different states of the ribosome itself.” More clear: “Thus, particular r-proteins may both sense the presence/absence of molecules in functional sites and also different states of the ribosome itself, much like molecular-scale neurons.”
Line 284: “In addition to sense and/or stabilize…” Should be: “In addition to sensing and/or stabilizing…”
Lines 287-289: “It has been proposed that with an average area of 200 Å2 and too tiny to be rationalized in terms of dimer stabilisation, these highly phylogenetically conserved interfaces have probably been selected during evolution to play a specific role in inter-protein communication. Their analysis revealed that most of them…” Should be: “With an average area of 200 A2, which is too tiny to be rationalized in terms of dimer stabilization, these highly phylogenetically conserved interfaces have probably been selected during evolution to play a specific role in inter-protein communication. These interfaces…”
Lines 299-301: “For example, the pathway such as the uL13-uL3 interface is universally conserved across the three domains of life, while others are replaced 300 by convergence in archaea and eukaryotes (fig. 6).” Should be: “For example, the 299 uL13-uL3 interface, which may function as a communication pathway, is universally conserved across the three domains of life, while other such interfaces are replaced by convergence in archaea and eukaryotes (fig. 6).”
Lines 370-376: (TYPOS UNDERLINED): “In summary, it is likely that communication pathways in ribosome may combine the multiple 370 repertoires of the allosteric mechanism that may exist in r-proteins (fig. 7B). In addition, it should be 371 noted that he ribosome provides a very particular electrostatic environment where the extensions are 372 immerged IMMERSED? in a sea of rRNA negatively charged phosphate groups. Neither electrostatic properties of 373 the proteins or the electron transfer rates in such a medium are well documented. However it is 374 known for example, that buried charged amino acids in proteins are much more charged that solvent 375 exposed ones [73]. It could be therefore BE expected that charge transport and the propagation of 376 electrostatic perturbations may have unusual properties in this particular electrostatic context.”
All these errors have been corrected in the new version (noticed in blue)
Line 434-442: “Even in simple organisms as tunicates having a few hundred of neurons, but close to the 434 vertebrates, several distinct sensori-systems are working together. For instance, in the adult ascidian 435 Ciona intestinalis, a wide range of innervation patterns connects two distinct systems the siphon and 436 dorsal nervous system [81]. The siphon system is a peripheral, sensori-motor system, controlling the 437 oral and atrial siphons, with eight siphon lobes and tentacles. It is responsible of food intake and 438 contains in particular peptidergic neurons. On the other hand, the dorsal system includes the cerebral 439 ganglion, responsible of absorption, digestion, excretion, reproduction, involving aCH, GLU, GABA 440 neuromodulators. In addition there exists a ventral nervous system, innervating the heart, the gill, 441 the stomach, and the endostyle (filter-feeding) [81]. Thus several networks, made by physiologically 442 different neurons, are interconnected and coordinated.” This paragraph needs a new introductory sentence that links it to the next paragraph: something along the lines of “One of the best analogies between ribosomal networks and simple neuronal networks is their modular structure. Even simple organisms such as tunicates, which are evolutionarily close to vertebrates, network several distinct, modular sensory systems together. For instance…”
We agree with this suggestion and we incorporate it in the text
Line 501: “to nervous systems at a nanoscale. These molecular systems are supposed PROPOSED to transfer and integrate…” [“SUPPOSED” HAS TWO MEANINGS IN COLLOQUIAL ENGLISH THAT CONFUSE THIS SENTENCE: PROPOSE SEEMS TO FIT YOUR MEANING BET
This has been corrected in the new version
Reviewer 2 Report
The hypothesis here suggested opens new perspectives in ribosome function, in evolution of complex systems and in biomimetic technological researches of nanoscale information transfer and processing. This would provide precious insights about the evolution of information processing in living organisms. It could also help to understand the complex behaviors of unicellular organisms that might use similar networks to integrate and respond external stimuli. Understanding the molecular mechanisms of information transmission and processing would constitute the basis for conceiving new computing nano-devices.
The only revision that I am suggesting consists in a brief discussion of how this way to see the ribosome might have importance in understanding the origin of ribosome itself and that of protein synthesis. After this minor revision, the manuscript might be accepted.
Author Response
Reviever 2
The hypothesis here suggested opens new perspectives in ribosome function, in evolution of complex systems and in biomimetic technological researches of nanoscale information transfer and processing. This would provide precious insights about the evolution of information processing in living organisms. It could also help to understand the complex behaviors of unicellular organisms that might use similar networks to integrate and respond external stimuli. Understanding the molecular mechanisms of information transmission and processing would constitute the basis for conceiving new computing nano-devices.
The only revision that I am suggesting consists in a brief discussion of how this way to see the ribosome might have importance in understanding the origin of ribosome itself and that of protein synthesis. After this minor revision, the manuscript might be accepted.
We have modified the previous title of the section 3.5.1 into ”Number of nodes, connectivity and evolution” and added a new figure (fig. 8) to describe briefly the evolution of the r-protein networks and the potential existence of universal r-network (that can be extracted from the comparison of bacterial and eukaryotic ribosome). These preliminaries results also converge with the concept of an autonomous ribosome as “a missing link in the evolution of life” [ref 37-39].